# Ecosystem Service Values as Related to Land Use and Land Cover Changes in Ethiopia: A Review

Muluberhan Biedemariam [1,2,*], Emiru Birhane [2] 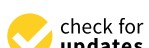, Biadgilgn Demissie [3], Tewodros Tadesse [4], Girmay Gebresamuel [2] and Solomon Habtu [2]

1  Department of Soil Resources and Watershed Management, College of Agriculture, Aksum University Shire Campus, Shire-Indasilassie P.O. Box 314, Tigray, Ethiopia
2  Department of Land Resources Management and Environmental Protection, College of Dryland Agriculture & Natural Resources, Mekelle University, Mekelle P.O. Box 231, Tigray, Ethiopia
3  Department of Geography and Environmental Studies, Mekelle University, Mekelle P.O. Box 231, Tigray, Ethiopia
4  Department of Agricultural and Resource Economics, College of Dryland Agriculture & Natural Resources, Mekelle University, Mekelle P.O. Box 231, Tigray, Ethiopia
*  Correspondence: muluberhan.biedemariam@mu.edu.et

**Abstract:** Humans worldwide depend on ecosystems and the services they provide. Land use and land cover change increasingly, influencing ecosystem values to the extent that the rate and direction of change occurred. The objective of this study was to review the link between changes in Land Use and Land Cover (LULC) and Ecosystem Service Value (ESV), with emphasis on mountainous landscapes in Ethiopia. The reviewers used the Preferred Reporting Items for Systematic Review and Meta-Analysis (PRISMA) guideline in the reviewing process. Area-specific and country-level studies showed that the ESV changed as the result of the LULC changes in the country. The change in land use in Ethiopia resulted not only in the loss of ESVs but also in the gain of ESVs depending on the type of man's activity. Negative change in LULC—especially the deterioration of land cover types such as forest land, shrub land and grass land—resulted in the loss of ESVs, whereas positive LULC change increased the value of ESVs. In Ethiopia, there is a loss of about USD 85 billion per year from the loss of ecosystem services. To save, improve and promote ESVs, land restoration and rehabilitation activities are important. The review provides insights into the need for and focus of future studies on LULC changes and the valuing of ESVs to understand the impact of changes in LULC on ESVs, considering existing and forecasted population increase in rapidly urbanizing areas.

**Keywords:** ecosystem services; Ethiopia; land use and land cover; change; linkage

## 1. Introduction

Human beings directly or indirectly depend on ecosystems and their services [1–6]. Several factors affect the healthiness of ecosystems globally. To Sutton et al. [2], the potential services that ecosystems provide to living things are limited by management and natural factors. LULC change due to anthropogenic activities and natural events is among the factors that bring changes in ecosystem services [6]. Understanding drivers to changes in LULC is crucial for enhanced ecosystem services [7]. The concern towards LULC change increases following the effect of these changes on biodiversity loss, soil degradation and the shrinkage of the role of the landscape to sustainably provide natural resources and ecosystem services [8]. The need to sustainably benefit from ecosystems and their services has captured the attention of researchers towards understanding the change in the value of ecosystem services through time [1,2,5,9–12].

However, there is a challenge in sustainably managing ecosystems that is mainly attributed to their inherent characteristics [13]. Changes in LULC increase the pressure on

the ecosystem services in different parts of the Ethiopian mountainous landscapes [1,5,9]. Similarly, the dynamics in LULC change the provision of ecosystem services in Nigeria [6]. Investigating the effect of land use change on ecosystem services is important for implementing suitable land uses and thereby improving ecosystem services [14]. Land use/land cover changes are among the potential determinant factors of ESVs, and thus, it is essential to understand the impact of future LULC changes on ESVs [1,15–17].

Understanding the future effects of LULC change on ecosystem services can give full information on the tradeoffs between possible choices of alternative land management options and uncertainties [1,17]. This can improve awareness of future effects of changes in LULC and provide insight into several land use management alternatives [16]. Moreover, understanding changes in ESVs as the result of change in LULC helps us in the design of land management alternatives for improving livelihoods and human well-being to influence development policies. The knowledge on the change in the value of ecosystem services is a crucial decision support tool for sustainable use of land resources [18]. However, the link between LULC change and ESV is not yet well known at national level in Ethiopia, especially in the mountainous landscape areas. This work reviewed ESV values as influenced by LULC change mainly in the mountainous areas of Ethiopia. The mountainous landscapes of Ethiopia have been under extensive and continuous deforestation and, consequently, dynamic change in LULC. This could have negatively affected the ESV of the landscapes, while in a few areas where proper landscape/natural resources management was practiced, the value of ecosystem services improved. This review tried to bring to the attention of policymakers and stakeholders the impact of LULC changes on the ecosystem services of mountainous landscapes in Ethiopia. Thus, the objective of the study was to understand the link between LULC changes and ESVs in the mountainous landscapes in Ethiopia to identify actions to save and promote ESV in the country.

## 2. Method

The literature review was conducted using the Preferred Reporting Items for Systematic Reviews and Meta-Analysis (PRISMA) guidelines [19] as shown in Figure 1. Initially, the literature was screened within four databases—Web of Science, Google Scholar, Scopus and Science Direct—for all articles in peer-reviewed journals written in the English language worldwide, with a particular emphasis on studies carried out in Ethiopia. The articles were searched using key words, phrases and names of authors. Cross-referencing was also used to find publications cited in the reviewed studies. Moreover, scientific journals, including Ecosystem Services, Science of the Total Environment and Global Environmental Change, were reviewed to find articles not identified through keywords, phrases and names of authors.

The reviewed literature consisted of studies from various disciplines that are either published articles or gray literature on land use and land cover; ecosystem services; land use and land cover change-driving factors; ecosystem service value change and driving factors; as well as impact of land use and land cover change on ecosystem service values. We used search terms to capture key words and phrases that are relevant to the topic of the study. The terms used in searching the literature included "land use and land cover", "change in land use and land cover", "ecosystem services", "ecosystem service value" "ecosystem service value in Ethiopia", and "environmental benefits in Ethiopia". In addition, greater emphasis was placed on land use and land cover impact on ecosystem services. The knowledge base of land use and land cover changes and ecosystem services is expanding at an increasing rate, and most of the articles included in this review were published in 2014 or later. Meta-analysis was not carried out because there are only a few studies that link LULC change and ESV in Ethiopia. The quantitative data of the reviewed articles was analyzed using Excel software.

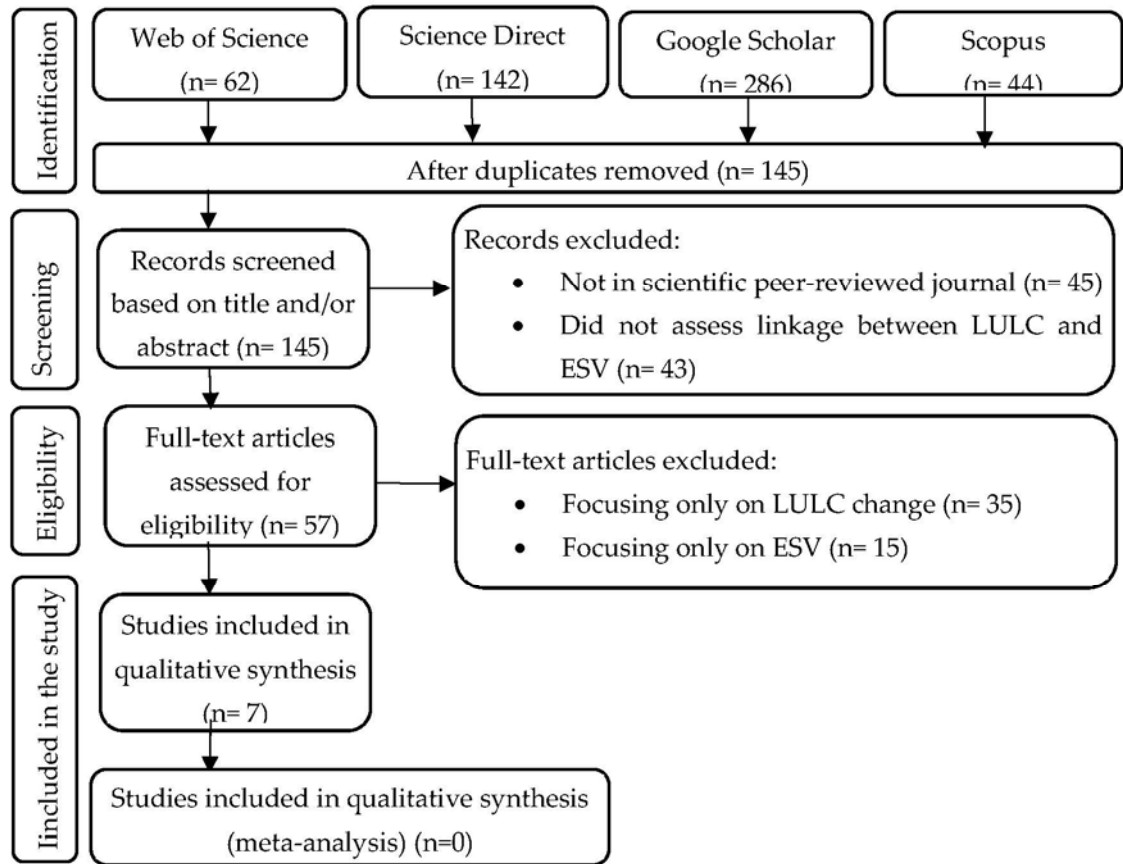

**Figure 1.** PRISMA flow diagram summarizing the search results and screening workflow used in this systematic review.

## 3. Ecosystem Service Value and Land Use/Land Cover

### 3.1. Land Use/Land Cover

At the global scale, the increase in research programs of land use can be traced back from the land use and land cover change project as a core part of international climate and environmental change research [20]. Land use and land cover are often used synonymously [21,22] although the correct meanings of these two phrases are quite distinct. Land use is the act of human beings on the land [23–25]. In other words, land use depicts how human beings are using the land for economic and social activities, such as agriculture, forestry, wildlife and recreation. Land cover indicates the physical land type, such as vegetation, water, urban infrastructure and bare soil. The global interplay of economic development and the conservation of biodiversity are reflected and determined by the change in LULC [26].

Studies towards LULC changes in developing countries increased with interest in facilitating sustainable land management through planning, monitoring and evaluation of various development programs (UNCED, 1992). According to Kleemann et al. [27] and Chang et al. [28], LULC change is the result of social and economic interventions as well as natural environment interactions. Land use and land cover change is the conversion of various land use types as the result of multifaceted interactions between man and the physical environment. Land use and land cover change has resulted in forest fragmentation, loss of biodiversity and land degradation [29,30]. The issue of land use and land cover change is one essential agenda item among global environmental changes and sustainable development [28,31] as it is a major global challenge [32].

*3.2. Ecosystem Service Value*

There are several widely accepted definitions of ecosystem and ecosystem services. Ecosystem—which is an interacting collection of plant and animal populations, together with their abiotic environment—is the central concept in ecology, and it can be defined at different scales from small to large and from local to global scope [33]. The UN Convention on Biological Diversity (CBD) defined ecosystems as "a dynamic complex of plant, animal and microorganism communities and their non-living environment interacting as a functional unit" [34]. One can easily understand from this definition that ecosystems are multifunctional. Ecosystem functions are the habitat, biological and system properties of ecosystems [35]. These functions provide various ecosystem services and benefits to people. Ecosystem services, on the other hand, are "the contributions of ecosystem structure and function (in combination with other inputs) to human well-being", as described by Burkhard and Maes [36]. These are the contributions made by ecosystems to benefits gained in human interventions, including economic, social, cultural and other activities [37]. These contributions show the role of the environment to people [38]. The concepts of ecosystem services and ecosystem goods and services are synonymous and used interchangeably [37].

Ecosystem services have a very important role in the function of nature and sustainable human well-being and survival [4,37,39,40]. These ecosystem functions and ecosystem goods that benefit human beings can come from either natural ecosystems or man-influenced ecosystems [41]. Ecosystem services in their natural state include fisheries and the collection of forest products and in their managed state include landscapes, such as crop systems of agroforestry, livestock keeping and aquaculture. Ecosystem benefit is the benefit that people get from ecosystem services [42]. The value of ecosystem service is the quantity of contribution of a given service from an ecosystem supporting the well-being of human beings [40]. Ecosystem services are basic to human existence on earth [5]. However, in a large part of Africa, persistent alteration and reduction of ecosystems is occurring at the expense of future generations' means of survival [4]. They are not the same spatially and temporally as they depend on the ecosystem type and status [43], especially in agriculture [44].

Ecosystem services are crucial for sustaining ecosystems' integrity though they are under stress because of the existing anthropogenic and climate change effects [5]. Though ecosystem services from mountain areas, for example, account for one-third of humanity, they are still exposed to both natural and human induced changes [45]. Several ecosystems provide unique services that cannot be substituted by others [9]. These can be through the improvement of human well-being and environmental quality [46]. Ecosystem services can enhance environmental quality in one way or another, for instance, through regulating ecosystem services, such as capturing air pollutants and providing fresh water [46] and climate and hydrological systems [6]. In addition, environmental quality can be improved through the provisioning of ecosystem services to obtain economic benefits from access to goods for subsistence or for the generation of wealth like feed, fuelwood and timber [1,6,39,47]. Furthermore, cultural services—such as aesthetics, contact with nature or recreation opportunities [48], as well as supporting services, including pollination and soil formation [6]—are also some of the services supplied by ecosystems.

## 4. Quantification Approaches of ESV and LULC Change

*4.1. Evaluation of LULC Changes*

There are several methods, like LULC changes, that are used for the evaluation and quantification of environmental phenomena. These range from ground-based measurements, which require area-wide and spatially explicit survey [22], to remote sensing [49] and modeling approaches that provide more information for large areas and simplify the monitoring purposes. The latter method is more reliable area-wide and quantifying at comparatively minimum costs, fast, frequent, and continuous observations for monitoring schemes. The ground-based technique is restricted to local scales due to high costs [22].

*4.2. Ecosystem Service Valuation*

Valuing ecosystem services is challenging [40] though many studies have been done on valuing ecosystem services by society [1,2,5,9–11,50]. Different techniques have been developed and used for the supply and demand assessment of ecosystem services; these techniques range from mapping and modeling to economic and non-economic valuation techniques [50]. According to Costanza et al. [3], the concern towards ecosystem services valuation increases from time to time following: (1) the development of ecosystem services valuation approach by [35]; (2) Gretchen Daily's edited book (1997); and (3) the Millennium Ecosystem Assessment [43]. Moreover, the recent study on The Economics of Ecosystems and Biodiversity (TEEB) (www.teebweb.org) has contributed to the knowledge and concern of the values of ecosystem services. Following the promotion of the concept of ecosystem services, countries have started to incorporate ecosystem services-based approaches into policy frameworks [36]. This can help to explain the opportunity costs of various programs and interventions.

There are different techniques for the valuation of environmental goods and services. Harrison et al. [50] developed a decision tree based on experiences from 27 case studies to help and guide the selection of comprehensive methods for ecosystem service valuation in a structured manner for different contexts. These supply and demand assessment techniques for ecosystem services are broadly categorized into biophysical modeling and sociocultural, monetary and integrative methods [50]. Valuing ecosystem services in terms of monetary values is essential to enhancing the awareness of users of ecosystem services and to presenting evidences for decisionmakers [6,51]. This would help the decisionmakers to sustainably manage environmental resources. The monetary methods are classified into nine categories: cost-effectiveness analysis; benefit–cost analysis; market price/exchange-based methods; revealed preference methods; stated preference methods; resource rent; simulated exchange; production/cost function; and value transfer methods [50].

Value transfer approach, more specifically the benefit transfer method, is a secondary valuation method. The benefit transfer method refers to the application of quantitative estimates of the value of ecosystem services from existing studies to another context [51]. It adapts established estimates of ecosystem service coefficients from primary valuation studies in one or more locations to other sites assumed to have related demographics, economic and ecological characteristics [17]. There are two ways of benefit transfer methods: function transfer and value transfer. The first approach is used to predict value coefficients for new study sites based on available data. On the other hand, the latter is used to transfer value from the original site to the new study site. This approach is important in the absence of site-specific valuation information. The benefit transfer technique has been used in several natural resources and environmental policies. Costanza et al. [35] used a simple benefit transfer method to understand the value of the world's 17 ecosystem services provided by 16 major biomes, and later, Costanza et al. [35] updated the estimate of the global ecosystem services values.

To calculate total ecosystem service values, several studies [1,5,6,17,52] applied Equation (1).

$$ESV = \sum(Ak * Vck) \tag{1}$$

where ESV = total ecosystem service value; Ak = the area of LULC type k in the study area in ha; and Vck = the value coefficient of LULC type k (USD/ha/year). These authors compared the land use classes of their study areas with the biomes proposed by [35]. Moreover, they used the most representative biomes as the proxy for the LULC category they used in their studies. The authors also used the updated monetary value calculated by [40]. Similarly, to calculate the value of individual ecosystem services function (ESVf) for LULC category 'k' in any landscape, researchers adopted Equation (2).

$$ESVf = \sum(Ak * VCfk) \tag{2}$$

where Ak = the area in ha of LULC type 'k' and VCfk = the value coefficient of function f (USD/ha/year) for LULC category, which can be obtained from Costanza et al. [34]. Furthermore, to understand the percent change of ESV across different periods of study years, [35,53] used Equation (3).

$$Percent\ of\ ESV\ change = \frac{(ESV recent\ year - ESV\ previous\ year)}{(ESV\ previous\ year)} * 100 \qquad (3)$$

However, when considering the limitation in the perfect matching of the biomes described and used by [35] and the LULC types as described above as well as the uncertainties of the value of the global coefficients, researchers used mechanisms for minimizing the uncertainties for overestimation or underestimation of ecosystem services. Various studies [1,5,54,55], for instance, used the adjusted value coefficients and conducted sensitivity analysis using standard economic analysis. This was to ensure the robustness and reasonability of their estimations of ESV [55]. The coefficient of sensitivity analysis was analyzed using the standard economic concept of elasticity, which is the percentage change in the output for a given percentage change in an input.

*4.3. Driving Factors to Changes in LULC and ESV*

Land use and land cover change dynamically exist on the globe due to human induced land conversions [2,5,12]. The interaction between natural, social, built and human capital is mandatory in order to produce different ecosystem services [3]. During this interaction to produce one service, another service can be affected. Findings indicate that the status of ecosystems and their potentials to supply services are changing from time to time [1,5,9,12,56,57]. The interest to gain short-term economic benefits at local level from agriculture has minimized human settlement challenges, increased space for resource extraction and thoroughly transformed the environment [58].

There are several LULC change drivers. Based on the consulted literature, the main causes for human-induced changes in LULC include population increase, economic development, social and biophysical factors, and the capacity of man to transform nature [12]. A study conducted in the semiarid river basin of India by Duraisamy et al. [59], for example, revealed that institutional factors, improving accessibility to agricultural water resources, and technological as well as economic factors were the key drivers to changes in LULC. The institutional factors included government policies and programs, legal frameworks, as well as mechanisms of governance. Similarly, population pressure, human interaction with the natural environment and change in economic development were recorded as the major LULC change driving factors in Ethiopia [2,5,30]. Conversion of land to produce crops, even recently, is increasing alarmingly [1,5,9,56]. The major driving factors to significant changes in ecosystem services are changes in LULC [1–5,9,14,59–62].

Changes in LULC may lead to variations in ecosystem service values [63,64]. Knowledge of the impacts of land use changes on ecosystem services, among other factors, is crucial in the era of global climate change, particularly to the sustainability of dryland ecosystems [65]. The status of ecosystems and the services they provide are affected by the decisions human beings made on land use management. For instance, Quintas-Soriano et al. [57] reported that, in Spanish drylands, land use management decisions, such as rapid expansion in greenhouse horticulture and urban intensification, minimized the regulation of ecosystem services. Undisturbed ecosystems and their ecosystem services are substantially affected by human interventions, such as agricultural activities, built-up areas, mining and settlements [1,5,6]. The effect, however, is not the same spatially and temporally [35,40]. Climate change is one of the factors that drive changes in LULC and ecosystem services [5]. It affects both local specific ecosystem services, such as pollination of agricultural crops, while also having the potential to mitigate climatic changes that are global in their nature. Climate change—through its direct or indirect effect on the alteration of hydrological processes, distribution of moisture–energy and the changing of carbon

dioxide concentrations—affects ecosystem services [66]. Climate change is threatening ecosystem services [67].

Furthermore, research findings confirmed that direct economic benefit though disasters to the natural environment are increasing the alteration of land uses, particularly natural ecosystems into agricultural lands [1,5,6,9,56]. Arowolo et al. [6] confirmed that the occurrence of agricultural land expansion is being manifested at the expense of forests and savannahs for the purpose of gaining short-term economic benefit. Apart from this, proper land management, land restoration and other land rehabilitation interventions through area exclosures enhances ecosystem services from deteriorated lands [7,8,10,68,69]. The study conducted by Biedemariam et al. [7] in Abreha-We-Atsibeha village in northern Ethiopia confirmed that the major drivers to the positive changes in vegetation coverage were land rehabilitation through soil and water conservation, and integration of trees as well as fruits on farms. Moreover, Dagnew et al. [69] and Mekuria et al. [8] confirmed the positive effect of restoring degraded lands on improving vegetation composition, carbon sequestration in vegetation and soil as well as improving both hydrological cycles and microclimate. Furthermore, Mekuria et al. [70] reported on the role that exclosures play in restoring degraded ecosystems through restoring soil fertility and native vegetation and ecosystem services.

## 5. Reviewed Studies from Ethiopia

### 5.1. Geographical Location of the Selected Studies

The studies selected for this review were performed in the representative mountainous highlands of Ethiopia from different locations (Figure 2). The locations of the studies are characterized by different climatic conditions (Table 1). The above-mentioned table shows the spatial distribution (geographic focus) and temporal aspects of these land use/land cover change impacts on ESV studies in Ethiopia. The reviewed studies spanned the mountainous highlands of Ethiopia.

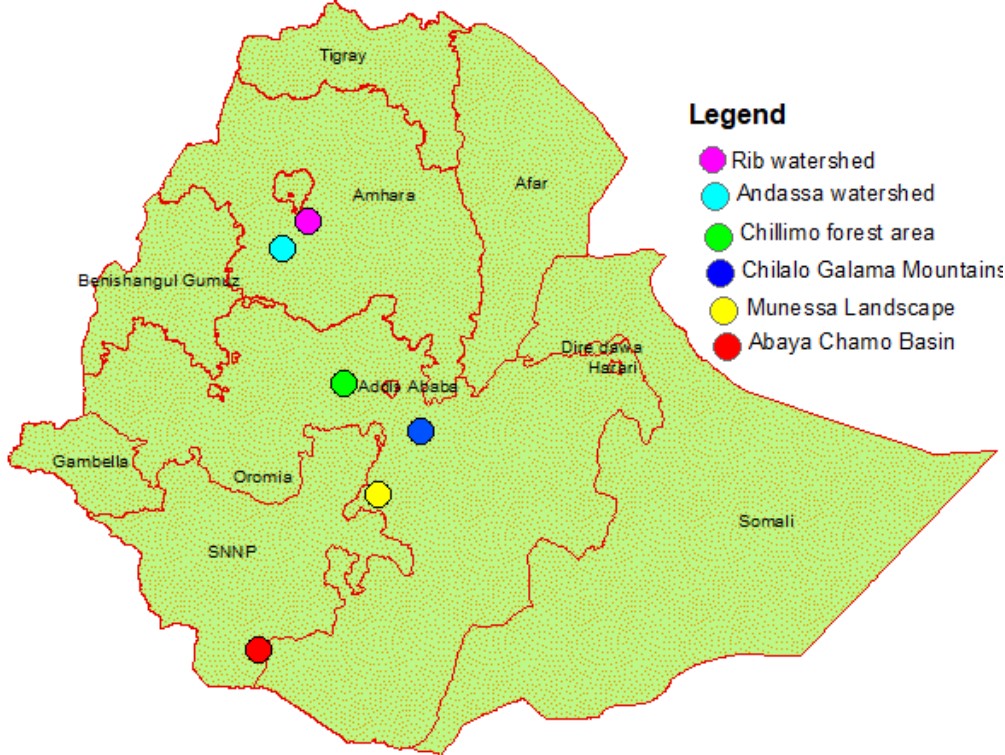

**Figure 2.** Map of Ethiopia showing the location of the studied areas reviewed in the study.

**Table 1.** Spatial and temporal distribution for the mountainous locations of the selected papers.

| Study Area of the Reviewed Paper | Region of the Study Area | Climatic Zone | Temporal Aspects |
|---|---|---|---|
| Andassa watershed in the Upper Blue Nile Basin of Ethiopia | Amhara regional state of Ethiopia | Highly sub-tropical (85.2%) with a small portion of temperate climate (14.8%) | 1985 to 2015 |
| Munessa-Shashemene landscape of the Ethiopian highlands | Oromia National Regional State of Ethiopia | Tropical dry Afromontane forest | 1973 to 2012 |
| Toke Kutaye district of West Shewa | Oromia National Regional State of Ethiopia | Part of the Central Highlands of Ethiopia | 1973 to 2014 |
| Rib watershed in the Upper Blue Nile Basin | Amhara regional state of Ethiopia, which stretches from mount Guna to the Eastern shore of lake Tana | Sub-tropical (64.4%), temperate (33.6%), alpine (2%) | 2000 to 2020 |
| Abay-Chamo basin in Southern Ethiopia | Southern Nations, Nationalities and Peoples' (SNNP) Region and, to a lesser extent, in the Oromia Region of Ethiopia | Humid climate in the mountainous highlands and a hot semiarid tropical climate in the lowlands | 1985 to 2010 |
| Chillimo forest area in Central highlands of Ethiopia | Oromia National Regional State of Ethiopia | Dry Afromontane forest | 1973 to 2015 |
| Chilalo-Galama Mountains | Oromia National Regional State of Ethiopia | Humid and sub-humid in the highlands and semiarid at some part of the study area | 1986 to 2021 |

*5.2. Changes in LULC*

Spatial and temporal empirical evidences from Ethiopia show that there has been considerable change in land use and land cover [1,5,7–9,16,29,56]. Gashaw et al. [9] carried out LULC change analysis to study the impacts of LULC changes on ecosystem service values in the Andassa watershed with an area of 587.6 square kilometers in the Upper Blue Nile basin of Ethiopia. The area is characterized by hilly topography, which shows a difference of altitude at small distances. The main economic activity in the watershed is agriculture. According to this study, there have been continued increments of cultivated land (22.5%) and built-up (1820%) areas and a reduction in forestland (41.52%), shrub land (44.97%) and grass land (36.11%) from 1985 to 2015 (Figure 3).

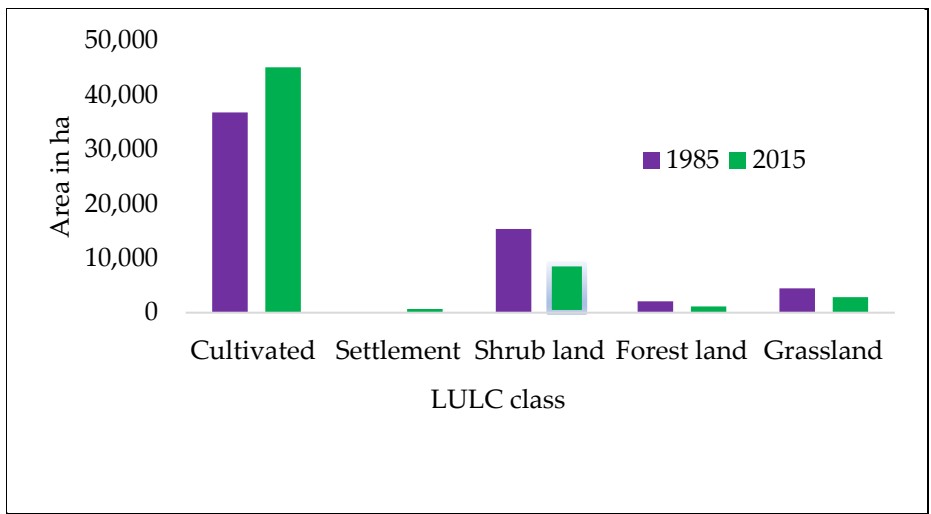

**Figure 3.** LULC changes from 1985 to 2015 in Andassa watershed, Ethiopia (Source of data: Gashaw et al. [9], but figure done by authors).

In a similar fashion, a study conducted by Tolessa et al. [5], in a typical dry Afromontane forest vegetation located in the highlands of Ethiopia, Chillimo forest area, reported

that shrub land, cultivated land and settlement increased by 437.7%, 9.1% and 6273.9%, respectively, from 1973 to 2015. Similarly, though there was no record for bare land in 1973, bare land coverage reached 739.1 hectares in 2015. A decrease by 54.2% of forest coverage from 1973 to 2015 is recorded (Figure 4). In terms of positive change, the highest percentage change in LULC occurred in settlement (6273.9%), and the lowest is recorded in cultivated land (9.1%). However, there was a shrinkage of forest land area coverage by 54.2%, and the area coverage decreases from 4263.1 hectares in 1973 to 1952 hectares in 2015 (see Supplementary Material).

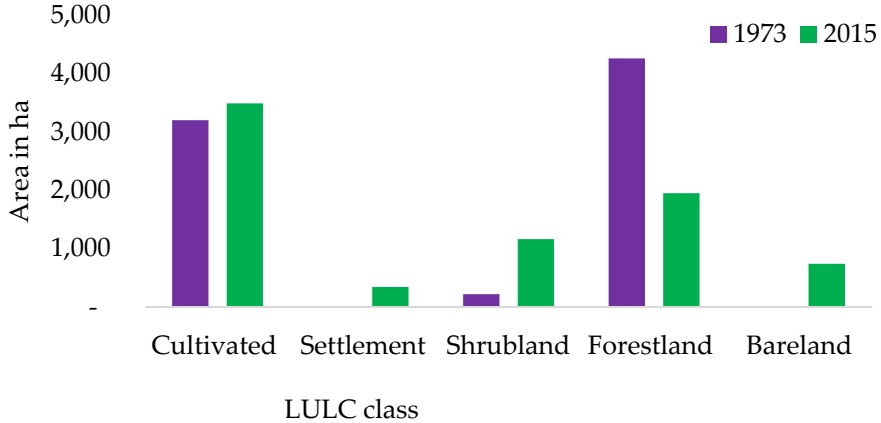

**Figure 4.** Changes in LULC from 1973 to 2015 in Chillimo forest, Ethiopia (Source of data: Tolessa et al. [5], but figure done by authors).

A study conducted in a 1091 square-kilometer area of the Munessa-Shashemene landscape of the Ethiopian highlands is characterized by a mixed farming system, confirming an increase in crop land, settlement, plantation forests, tree patches and bare lands [1]. Similarly, a reduction in area coverage of woodlands, natural forests, grasslands and water body was recorded in the last four decades [1]. In 1973, 42% of the land was covered with grasses followed by natural forests (21%), cropland (13%), woodlands (11%), water body (10%), tree patches (2%) and settlement (0.42%). The area under crop fields covered about 48.69% of the total area of the landscape in 2012. In this study, the highest percentage of change in land use and land cover happened in bare land, which is 414.6% followed by crop land with 272.8% and settlement with 261.3% as well as the lowest change occurring in tree patches (Figure 5). However, in reduction in area of coverage perspectives, the highest change is recorded in natural forest, which decreased from 21,726 hectares to 9588 hectares (see Supplementary material).

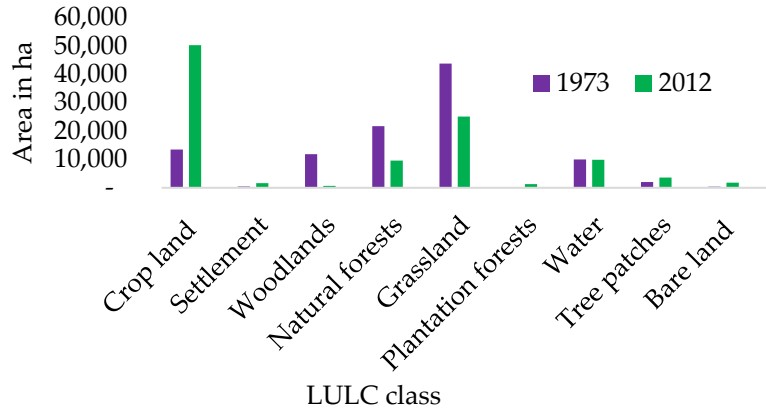

**Figure 5.** Changes in LULC from 1973 to 2012 in Munessa-Shashemene, Ethiopia (Source of data: Kindu et al. [1], but figure done by authors).

Another study carried out in 1972, 697.2 hectares of Toke Kutaye area, which is characterized by a mixed farming system and with an estimated population density of 198 persons per square kilometer, reported an increase in cultivated land and settlement while a decrease in shrub–bush land, grass land and forest land as well as bare land [20]. In 1973, the area coverage of cultivated land was 13,424.5 hectares while in 2014 its area coverage was 43,286.4 hectares (Figure 6). Similarly, the area coverage of settlement increased from 1202.13 hectares to 6869.82 hectares from 1973 to 2014, respectively (see Supplementary Material). While forest land, grass land and bare land classes declined by 83.23%, 46.64% and 37.35%, respectively, from 1973 to 2014.

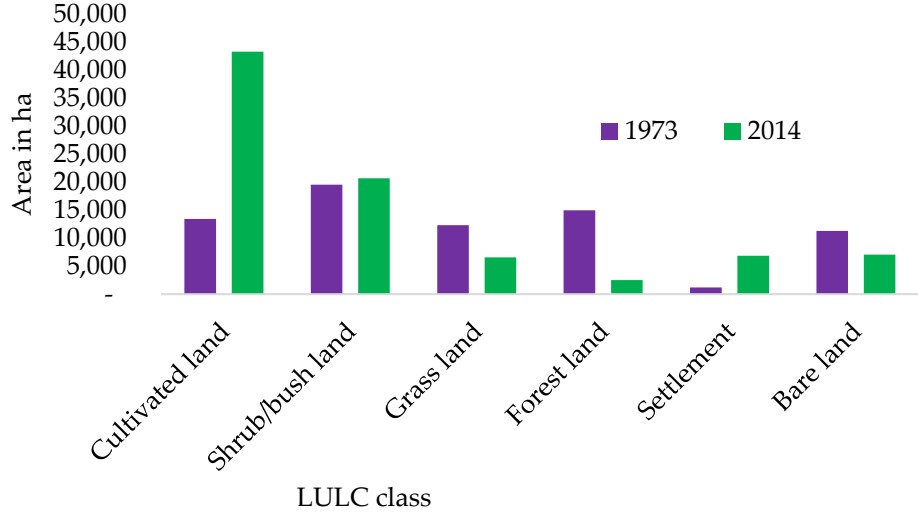

**Figure 6.** Changes in LULC from 1973 to 2014 in Toke Kutaye, Ethiopia (Source of data: Tolessa et al. [56], but figure done by authors).

A study conducted by Biratu et al. [22], in an area of 10,074 square kilometers in the Great Rift Valley of which extended up to the Chilalo mountain, it was also noticed that the highest change was detected on cultivated land, which resulted in a 7.3% change throughout the study periods (1986–2020). Built ups and bare lands had been increased positively during this study period. On the other spectrum, the least change was detected in the reduction of water body by 0.1% (Figure 7). Like the findings of the other studies reviewed in this work, shrub–bush land and the rest of the other land use and land cover classes decreased from 1986 to 2021 (Figure 7).

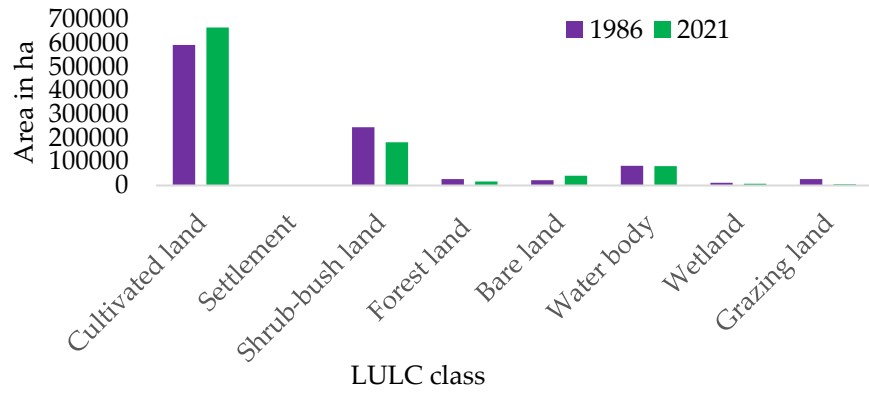

**Figure 7.** Changes in LULC from 1986 to 2021 in Chilalo mountain, Ethiopia. (Source of data: Biratu et al. [71], but figure done by authors).

Of the identified LULC classes in the Rib watershed during 2000–2020 (Figure 8) the highest positive percentage change was detected on settlement (136.5%) though this land

cover class is covering the second least proportion of that landscape. During this period, the lowest positive percentage change was detected on cultivated land (23%)—a class covering the maximum proportion of the total area of the watershed. Contrarily, the maximum negative percentage change was detected on forest (46.5%), and the minimum negative percentage change was detected on grassland (41.5%).

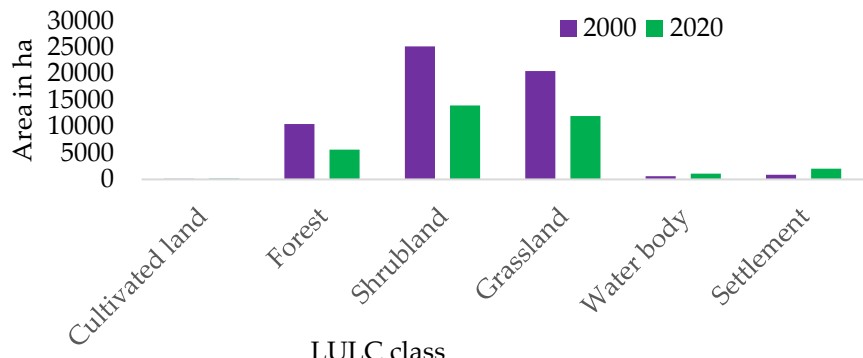

**Figure 8.** Changes in LULC from 1986 to 2021 in Rib watershed, Ethiopia. (Source of data: Anley et al., 2022 [72], but figure done by authors).

In 25 years (1985–2010), the Abaya-Chamo Basin had shown a rapid increase in the area coverage of arable land and its coverage in the total area increased from 23.3% in 1985 to 37% in 2010 (Figure 9). Whereas, shrub land, natural grassland and heterogeneous agricultural areas had reduced their contributions to the total area of the basin from 23.4%, 16.7% and 13.2% in 1985 to 16.7%, 11.2% and 11.3% in 2010, respectively.

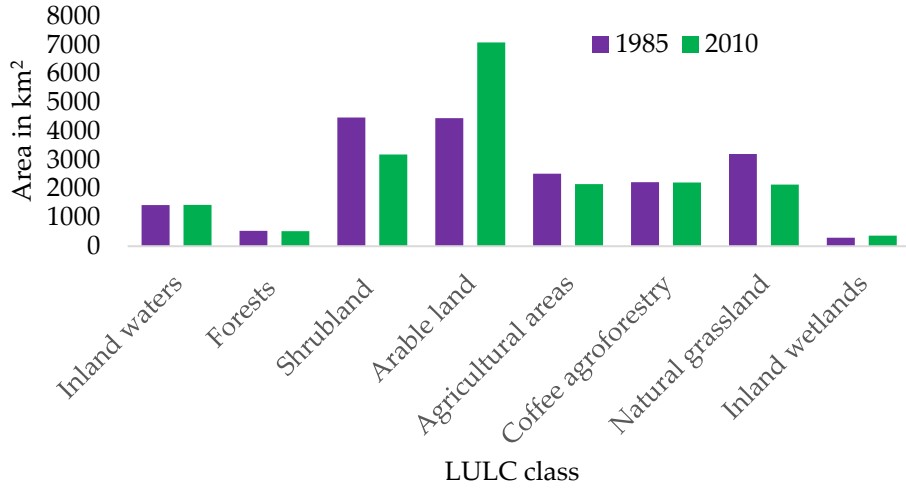

**Figure 9.** Changes in LULC from 1985 to 2010 in Abaya-Chamo Basin, Ethiopia. (Source of data: Woldeyohannes et al., 2020 [73], but figure done by authors).

### 5.3. Ecosystem Service Value Changes

In the Andassa watershed of 587.6 square kilometers area, there occurred a gain of USD 1.86 million from cultivated land from 1985 to 2015. While there was a loss of USD 6.3 million, USD 920,000 and USD 4.7 million during the same period from shrub land, forest land and grass lands, respectively. According to this study, a 22.5% increase in area of cultivated land contributed to an increase in ESV of USD 1.86 million. However, the increase in area of cultivated land was compensated by 41.52% of shrub land, 44.97% of forestland and 36.11% of grassland decreases that together resulted in a loss of USD 7.69 million ESVs. In this watershed, there was a loss of USD 5.83 million ESVs in the last 30 years. This indicated that the change in LULC of the watershed led to the loss of ESVs.

Unlike these changes, there was not any change in the ESV of the settlement land cover class of the watershed (Figure 10).

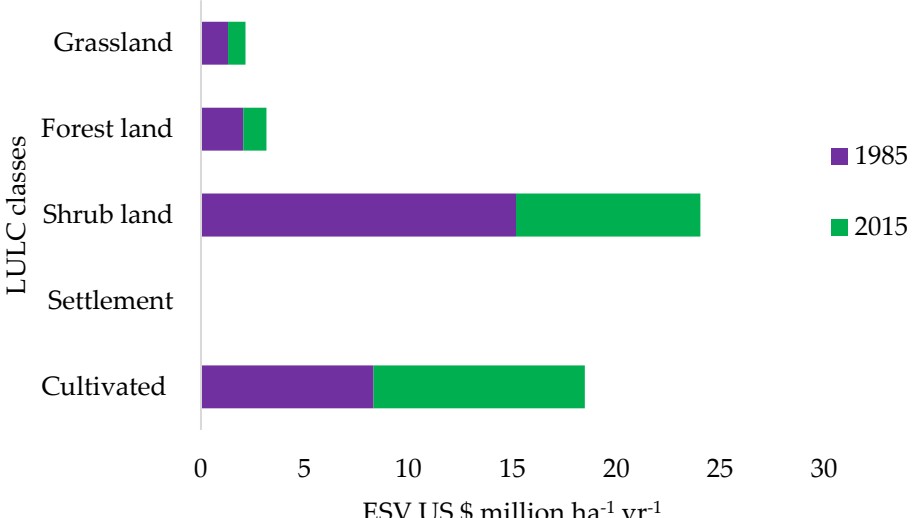

**Figure 10.** ESV changes from 1985 to 2015 in Andassa watershed, Ethiopia (Source of data: Gashaw et al. [9], but figure done by authors).

From 1973 to 2015, an increase in ESV of USD 31,000 is estimated from cultivated land of the Chillimo forest area. Whereas, the ESV of forest land and shrub land in the area from 1973 to 2015 is decreased by USD 4.64 million and USD 80,000, respectively, while the other LULC classes (settlement and bare land) did not show any change in terms of ESVs during the last 42 years in the area. Cumulatively, at the landscape level, there was a loss of USD 4.52 million during the last 42 years in the Chillimo forest area (Figure 11). This reduction in ESV of the landscape was caused by a 54.2% decrease in the area coverage of forest land.

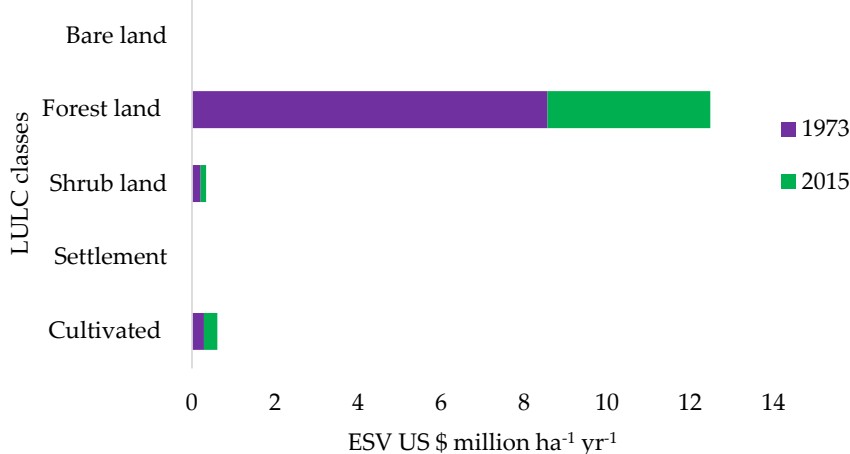

**Figure 11.** ESVs from 1973 to 2015 in Chillimo forest, Ethiopia (Source of data: Tolessa et al. [5], but figure done by authors).

A study conducted by Kindu et al. [1] in a 1091 square-kilometer area characterized by a mixed farming system confirmed a gain of USD $3.4 \times 10^6$ from crop land, USD $2.6 \times 10^6$ from plantation forests and $4 \times 10^6$ from tree patches from 1973 to 2012, respectively. While there was a loss of USD $24.3 \times 10^6$ from natural forests, USD $22.5 \times 10^6$ from woodlands, $4.6 \times 10^6$ from grasslands and $9 \times 10^5$ during the last 40 years, respectively (Figure 12). When looking at the difference between the gain and loss of total ESV during the last 40 years in the total landscape, the loss is greater than the gain by more than 8-fold, which resulted in a net loss of USD 45 million ESV caused by the dynamics in the LULC between

1973 and 2012. The damage of ESV totals USD 52.3 million while the total gain in ESV in the same period is USD 6.4 million. Of the total loss in ESV of the landscape, change in natural forests was responsible for the largest loss, accounting for 46%, followed by woodland with a 43% contribution, while the water bodies and grass land decrease accounted for about 2% and 9%, respectively. The positive change in total ESVs is contributed by the change in cropland (53.13%) and plantation forests (40.63%).

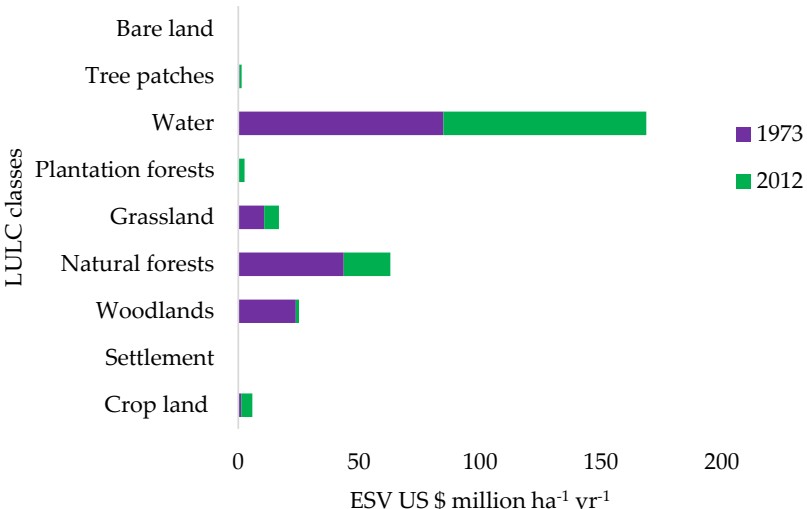

**Figure 12.** ESVs from 1973 to 2012 in Munessa-Shashemene, Ethiopia (Source of data: Kindu et al. [1], but figure done by authors).

Analysis of the total ESVs for the Tokye Kutaye area shows a deterioration of USD 36.29 million ESVs from USD 53 million in 1973 to USD 16.71 million in 2014 [56]. The highest change in ESV is recorded in shrub–bush land that is USD 12.7 million, while there is no ESV change in both settlement and bare land classes of the area.

The ESV of forestland, shrub–bush land and grassland declined from USD 30.1 million, USD 12.7 million and USD 2.85 million in 1973 to USD 5.03 million, USD 6.2 million and USD 1.35 million in 2014, respectively (see Supplementary Material). During the same period, the ESV of cultivated land increased by USD $2.74 \times 10^6$; however, this is much less than the loss of the ESV (Figure 13).

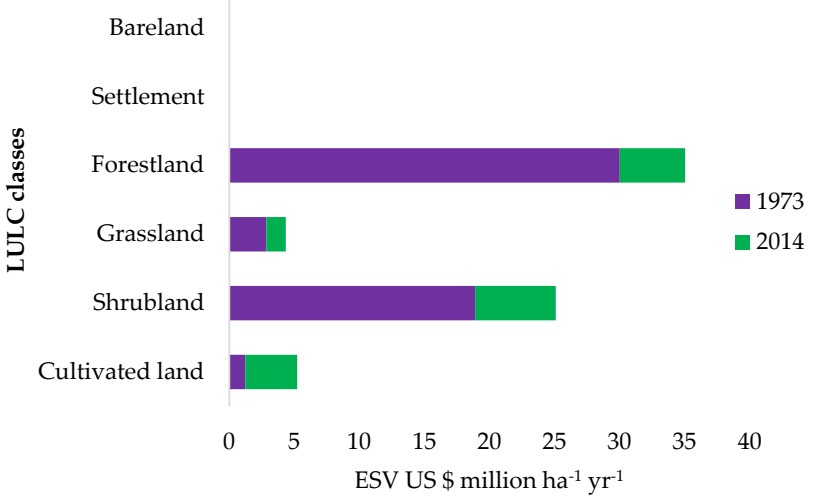

**Figure 13.** ESVs from 1973 to 2014 in Toke Kutaye, Ethiopia (Source of data: Tolessa et al. [56], but figure done by authors).

The total ESV in Chilalo mountainous areas decreased by 5.9% (USD 58.8 million) from 1986 to 2021 (Figure 14). This reduction in ESV was contributed by the wetland areas, shrub–bush land, natural forests, waterbody and grazing land LULC classes in their order of reduction. Grazing land's ESV decreased during the mentioned period by 81.1%. In contrast to this, the ESV of cultivated land increased by USD $16.5 \times 10^6$.

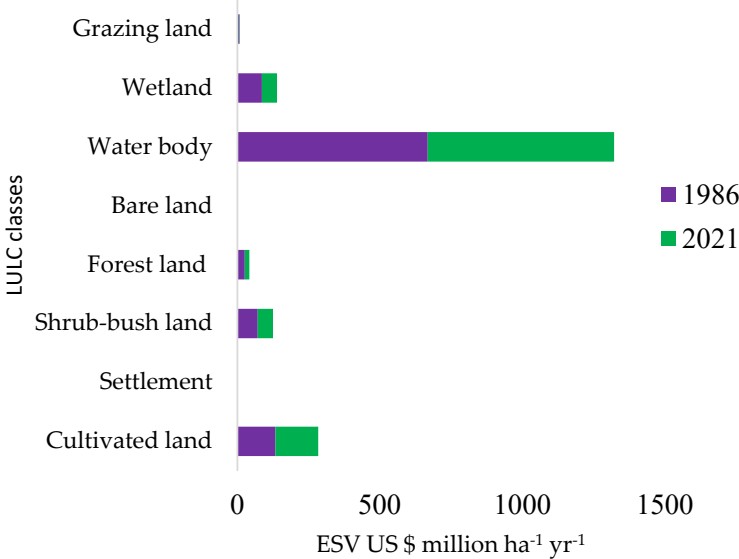

**Figure 14.** ESV from 1986 to 2020 in Chilalo mountain, Ethiopia (Source of data: Biratu et al. [71], but figure done by authors).

From 2000 to 2020, in the Rib watershed of the Upper Blue Nile basin of Ethiopia, the total ecosystem service value decreased by 13.5%, and this loss mainly resulted as the consequence of the reduction in the ESV of shrubland and forest. The only LULC that recorded a positive ESV contribution throughout the study periods was cultivated land (Figure 15).

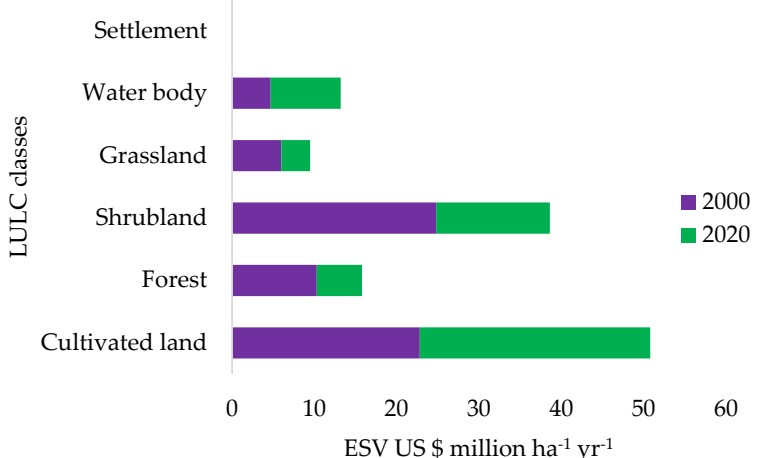

**Figure 15.** ESV from 2000 to 2020 in Rib watershed, Ethiopia (Source of data: Anley et al. [72], but figure done by authors).

In the Abaya-Chamo basin of Southern Ethiopia, land cover classes of arable land, inland wetlands, built-up areas and inland waters were among the classes that positively contributed to the total ESV of the landscape (Figure 16). The total ESV of the landscape increased by 2.67% within a 25-year span. A 59.2% positive change in arable land resulted in USD 1466.9 million in the Abaya-Chamo basin of Ethiopia, which mainly contributed to

the overall enhancement of net USD 331.1 million ESV of the basin from 1985 to 2010 while the shrub land class of the basin decreased by USD 694.5 million during the same period.

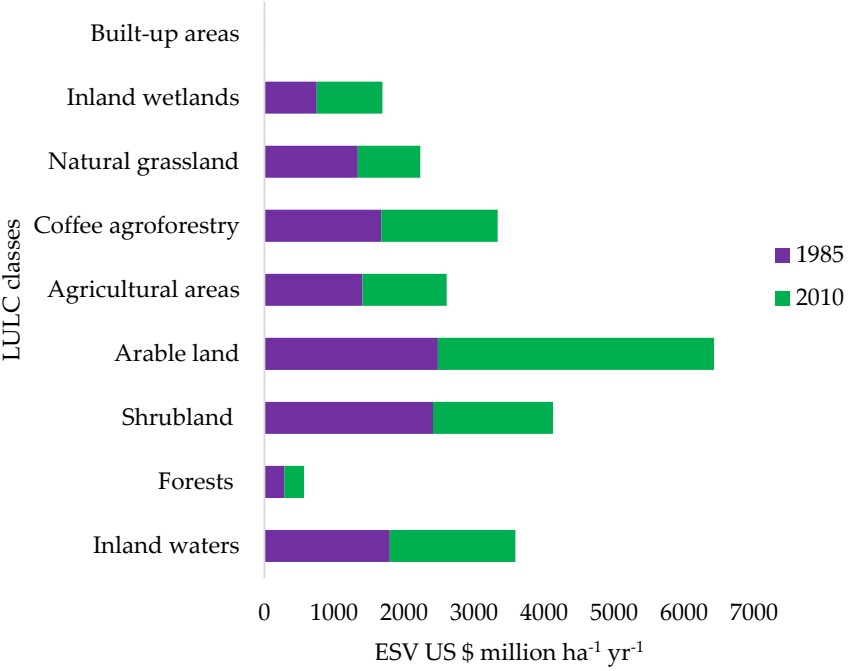

**Figure 16.** ESV from 2000 to 2020 in Abaya-Chamo Basin, Ethiopia (Source of data: Woldeyohannes et al. [73], but figure done by authors).

## 6. Discussion

The relationships between LULC changes and the value of ecosystem services with a special focus on the Ethiopian mountainous landscapes have been identified through the analysis. The area-specific studies reviewed in this work confirmed change in ESVs as the result of LULC changes, despite spatial and temporal differences in the studies. The analysis of the effect of LULC changes on the ESV showed a reduction in the services mainly related to the reduction in the areas of forest and shrub–bush land. This could have implications for people and biodiversity. There was a dramatic increase in cultivated/crop land, barren land and settlements during the study periods in the study areas. This suggests that, as time increased, the demand for cultivated land and shelter increased. This might have increased with the increase in population and food demand and the consequent expansion in industries and factories to meet the demand of the increase in population. Cultivated/crop land increase in the studied areas in Ethiopia ascribed to the land policy, the adoption of Agricultural Development Led Industrialization (ADLI) and agricultural development policy. In the 1990s, the Federal Democratic Republic of Ethiopia institutionalized ADLI as the main engine of the country's economy [74]. Consequently, farmland expansion has been one of the key factors that contributed to the agricultural growth in Ethiopia [75]. Besides, the land ownership policy of the country stipulated in the constitution, which stated land as the property of the state [76], allowed rural people to convert woodland forest, shrub–bush land and grazing land into cultivated/crop land. Further, the agricultural development policy in the country, failing to meet subsistence needs with undeveloped farming systems [77–79] and large-scale expansion of traditional based commercial farming in unused land [80], increased cultivated/crop land. The increase in cultivated/crop land could be compensated by the decrease in forest area, shrub–bush land and grass/grazing land coverage of the selected study areas.

The development of techniques for ecosystem services' mapping and quantification, to know their future situation through models, increased works of LULC change scenario analysis [16] and their impact on ESV. This enabled the prediction of the impact of various

LULC change scenarios on ecosystem services based on past trends and scenarios on planning alternatives [1,15] as well as on the long-term strategies [16]. From the reviewed literature, despite different contexts in different regions regarding LULC changes and the existence of ecosystem services, several challenges and problems are resulting from the impact of LULC changes on ESVs. As the mountainous landscapes of Ethiopia are dominated by agriculture and human settlements, their ecosystem services are affected by LULC changes that, in turn, are derived by the interaction of several activities [5]. Understanding several services of ecosystems quantitatively is crucial for awareness raising and decisionmaking processes at all levels [5,9]. The consequence of land use change on ecosystem services can be positive or negative [14]. The change in ecosystem services is accounted for indirectly from the impact of land degradation on ecosystem functions, and there is a significant reduction in the value of the services [2].

Our analysis of the different studies was in line with the studies conducted at national levels. For instance, the land degradation in Ethiopia, as the result of LULC change among other factors, has affected a loss of USD 85,419,048,953 (17.7% of the total terrestrial ESVs) [2]. Similarly, Song and Deng [81] assessed LULC change and ecosystem services provision in China and found that 1% of land conversion led to an average change in 0.1% ESV from 2000–2008. Sutton et al. [2] assessed the impact of land degradation on ESVs globally and found a total loss of USD 6.3 trillion ESV per year. By looking at the global GDP of the 2010s, which was USD 63 trillion, and the contribution of agriculture (which was about 2.8%), Sutton et al. [2] reported that the loss in ESV is more than threefold the contribution of agriculture to GDP. This reflected that the market value of agricultural products cannot describe fully the economics of land deterioration [2]. Costanza et al. [40] estimated USD 4.3–20.2 trillion per year loss of ecosystem services as the result of land use changes worldwide. With the assumption of 100% of the ecosystem functioning, about an annual USD 20 trillion was lost globally as the result of the impact of land cover change alone for the last 15 years [40]. The value in the loss of ecosystem services is higher than the market value of agricultural production globally.

From the analysis of all the area-specific empirical studies in the mountainous landscapes of Ethiopia, this study showed that the changes in LULC classes and ESVs are in both directions. Several studies conducted at different spatial and temporal scales showed the effect of LULC change on ESVs in two ways. That is, in some areas, there existed a deterioration of ESVs, while in other areas, improvements have been confirmed. On one hand, studies showed the changes in LULC and, consequently, the loss of value of certain ESVs to be the result of agricultural encroachment to forested areas [5,9,56]. On the other hand, plantation forests increased ESVs [2]. Moreover, proper land management, land restoration and other land rehabilitation interventions through area exclosures, soil and water conservation activities enhanced ecosystem services from deteriorated lands [7,10,69]. The studies indicate that, as the number of years increased, the change in LULC classes continued. The reason would be that, as the number of years increased, the area of cultivated land increased; this would suggest that the demand for cultivated land increased as the number of people increased in the study areas. These findings indicate that, in the future, the demand for cultivated land and settlement will increase, which in turn, will create additional LULC changes. The change would be from the remaining forested lands, shrub–bush lands and from forest plantations. This might result in the degradation of ecosystems and the services humans obtain in rural areas.

According to the United Nations Department of Economic and Social Affairs (2017), Ethiopia is ranked fifth out of the nine countries expected to contribute to one-half of the world's population growth from 2017 to 2050. The population growth of Ethiopia is seen as a threat for environmental challenges on land resources of the country [10]. Considering the fact that population growth is among the key LULC change-driving forces, the pressure of population growth in changing the LULC, if not properly and wisely managed, will increase and consequently will deteriorate the value of ecosystem services in Ethiopia. Therefore, proper management of population growth, in relation to the ecosystem functioning, would

result in comparable LULC change. Besides, the urban population of Ethiopia, and Tigray in particular, is rapidly increasing. The growth in urban population, if not properly managed, could affect the ecosystems of urban areas. Thus, presenting scientific evidences on the effect of LULC change on ESVs in urban landscapes could enable us to plan for sustainable urban planning with consideration for urban ecosystems.

## 7. Conclusions

Ecosystems and the multiple services they provide are crucial to the survival of man on earth. The present systematic review was done to understand LULC change, ESV and their relationships. Change in LULC has been creating challenges in ESVs. Though the changes in the values of ecosystem services vary from country to country, there is a record on the loss of ESVs worldwide and in Ethiopia in particular. The driving forces for LULC change vary with regions and countries, indicating the need for further understanding of LULC dynamics on country- and local-level scales. The main driving force for the changes recorded in all the study areas of this study is the interaction of man on the environment. Though studies conducted in Ethiopia in relation to LULC changes and ecosystem services did not cover the whole country, results of local-level investigations have produced important information that can be used for future land management and land use planning activities.

All the consulted studies showed a positive change in the land use of cultivated land and an increase in ESV of this land use class. Similarly, positive change in plantation forest area increased ecosystem service value while a reduction in the area of natural forest coverage resulted in a decline of ESV. However, the change in ESV of cultivated land is smaller than the loss of ESV as the result of forest area and shrub–bush land decline. Thus, it is important to increase efforts for land restoration and rehabilitation, such as tree plantations, and to minimize natural forest degradation to enhance ESVs. There are more works of research on the impact of LULC change on ecosystem services based on past trends than there are studies that predict impacts on existing and planned programs and strategies. Considering this gap and in order to improve the human well-being of the present and future generations, efforts should be increased to predict impact of LULC changes on ESVs to act and manage the natural resources and their ecosystems accordingly.

Moreover, all the studies reviewed from Ethiopia were conducted in rural areas though settlements affect ecosystems and their service values. Hence, it could also be crucial to study the association between urban LULC dynamics and ESVs in urban landscapes of Ethiopia to raise the awareness of urban planners and decisionmakers. Thus, to sustainably manage and improve existing ecosystems, knowledge—on the existing status of the LULC and empirical evidences on the effect of changes in LULC on ecosystem services at different spatial levels and temporal aspects of both rural and urban areas—is crucial.

Furthermore, scenario analysis on the LULC change impact on ecosystem services considering the existing and forecasted population of the country is essential. Hence, to improve understanding on the LULC change impact on the changes in the ESVs, mainstreaming the evaluation of LULC change with ESV is required following state-of-the-art procedures. It is therefore essential that academic and research institutions understand the importance of building their capacity in terms of the valuing of ecosystem services.

**Supplementary Materials:** The following supporting information can be downloaded at http://www.mdpi.com/xxx/s1, Table S1: Land use and land cover changes with ecosystem service values of different areas in Ethiopia.

**Author Contributions:** M.B. designed the research for this paper and took the lead in collecting the data, analyzing and writing the paper. B.D., E.B., G.G., S.H. and T.T. commented and edited the paper. All authors provided invaluable comments that helped shape the research, analysis and manuscript. All authors have read and agreed to the published version of the manuscript.

**Funding:** This research was funded by UK Research & Innovation (UKRI) through the Global Challenges Research Fund (GCRF) programme, Grant Ref: ES/P011306, under the project Social and Environmental Tradeoffs in African Agriculture (SENTINEL), led by the International Institute for Environment & Development (IIED) in part implemented by the Regional Universities Forum for Capacity Building in Agriculture (RUFORUM). The ABC was funded by UK Research & Innovation (UKRI) through the Global Challenges Research Fund (GCRF) program, Grant Ref: ES/P011306, under the project Social and Environmental Tradeoffs in African Agriculture (SENTINEL), led by the International Institute for Environment & Development (IIED) in part implemented by the Regional Universities Forum for Capacity Building in Agriculture (RUFORUM).

**Data Availability Statement:** Not applicable.

**Conflicts of Interest:** The authors declare no conflict of interest.

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
