# Peer review of "Ecosystem Service Values as Related to Land Use and Land Cover Changes in Ethiopia: A Review"

_land, doi:10.3390/land11122212_

Round 1

Reviewer 1 Report (Previous Reviewer 1)

A location map needs to be given

How different locations used in the study are related

The data used is from other sources. What is the integrity of data made by the earlier worker? What is the similarity in data preparation by these workers? All need to be discussed in the methodology.

What is your analysis of data other than presentation in bar charts? You should analyze data in light of population pressure, natural setup, physiography, etc to discuss the rate of changes.

Some areas show no changes in water bodies even with drastic changes in the forest, justify such observation.

waterbodies shown in data are man-made or natural, need to be given.

Some areas show no changes in forest areas whereas some show built-up development. Discuss these significant driving forces in LULC changes and their impacts.

Analyze and compare all studies together and focus on why abnormal changes happen in some areas and not in others with their implication.

Author Response

Please  find attached here with the  response to the comments.

Reviewer 2 Report (Previous Reviewer 2)

General comments

This paper reviewed the link between changes in land use and land cover and ecosystem service values with emphasis to mountainous landscapes in Ethiopia by using the Preferred Reporting Items for Systematic Review and Meta-Analysis (PRISMA), which is of great significance for understanding the impact of changes in LULC on ESVs. However, there are some problems, which must be solved before it is considered for publication.

Specific comments

1.Uniform chart format. Such as column chart interval, thickness, position of the legend, etc.

2.The introduction of the method in this study is inadequate. Does software are used for data processing? if so, please indicate in the text.

3.Whether the keyword search is insufficient? For example, Land use change is often used to study without distinguishing the land use and land cover change.

4.In Figure 1, is meta-analysis (n=0) normal and how to explain it?

5.There are many methods listed in Quantification Approaches of ESV, but only the value transfer approach is detailed, and the advantages and disadvantages of each method are not fully analyzed and discussed.

6. Unified data units and use the commas in numbers correctly. Such as 6.3million and 4,700,000 in line 425, km2 and ha, etc.

7. The part of Reviewed studies from Ethiopia had incomplete results, listing only seven available studies and lacking analysis.

8. The Discussion repeatedly listed the Ecosystem Service Value Changes in 5.3, which should be described briefly.

9. Explain the differences between this review and other relevant research reviews, and refine the innovation and necessity of this review.

Author Response

Please kindly find attached here with the response to the comments.

This manuscript is a resubmission of an earlier submission. The following is a list of the peer review reports and author responses from that submission.

Round 1

Reviewer 1 Report

The manuscript entitled “Ecosystem Service Values as Related to Land Use and Land Cover Changes in Ethiopia: A review” presents a critical review of different methodologies of ecosystem service and land use changes. A detailed review of the manuscript is described below.

The abstract need more qualitative information derived from the study.

There is no adequate discussion about land use patterns and transformation in the study area.

What are the different numerical models that can be used for a similar study, and which method is more significant than others?

There is no source of data for figures. Is it done by the authors,

The manuscript required restructuring based on the journal format, however, the authors nicely presented the manuscript, but as per the journal format result and discussion should be separated.

the manuscript requires more citations in order to justify the review, which should be from a global perspective to a regional one.

There is a certain confusion in the manuscript flow.

I recommend a minor revision in the version.

Reviewer 2 Report

The topic about "LUCC and ESVs" is not fresh. I suppose this paper has two potential highlights: 1. using a meta-analysis method; 2. focusing on mountainous areas. However, a major revision is needed.

1. I didn't see any results or  elaborations related to mountainous areas of Ethiopia  after the Introduction. I suggest you to focus on "Moutainous area of Ethiopia" thoughout the whole paper, thus finding something interesting about the spcific landscape. Otherwise, this paper would lose the only innovation.

2. Mapping your study area explicitly to readers. At least, you have to show where the mountainous landscape of Ethiopia is, and what the latest land cover is. 

3. The linkage between LUCC and ESVs in Ethiopia is still not clear, which is the key problem to be solved in this paper. ESVs change along with land use change is just a superficial phenomenon. What is the reason for his phenomenon in Ethiopia? Maybe, this study needs to furthermore review more researches on "trade-offs between ecosystem services under the land use change" .

Reviewer 3 Report

The goal of this review was to examine the relationship between land use and land cover changes and ecosystem service values ESV in Ethiopian mountainous landscapes. This is an interesting topic, though very narrow scope. The authors also conducted a meta-analysis based on the findings of a few studies that found a link between land use and land cover change and ESV changes in Ethiopia.

The article could be published in Land after substantial changes. Unfortunately, this is not a meta-analysis (it looks like a report), authors should look at published meta-analysis examples.

www.sciencedirect.com/science/article/pii/S0921800916309168

https://www.sciencedirect.com/science/article/pii/S2212041621000206

https://www.cambridge.org/core/journals/agricultural-and-resource-economics-review/article/using-metaanalysis-for-largescale-ecosystem-service-valuation-progress-prospects-and-challenges/D35E2C7D3BC167026FECA3CC29B99921

the methodology also needs substantial revisions, it is not clear how the articles were selected, and the temporal aspects.

Other minor issues:

Abstract: Spell out ESV

Page5 line 200-201: The articles were searched using key words, phrases, name of authors. Name of authors?

6. Include Results. How many papers have you found?

Figures must be improved.

Include tables, e.g., results from the meta-analysis.